# Development of a Formability Prediction Model for Aluminium Sandwich Panels with Polymer Core

**DOI:** 10.3390/ma15124140

**Published:** 2022-06-10

**Authors:** Xiaochuan Liu, Bozhou Di, Xiangnan Yu, Heli Liu, Saksham Dhawan, Denis J. Politis, Mateusz Kopec, Liliang Wang

**Affiliations:** 1Department of Mechanical Engineering, Imperial College London, London SW7 2AZ, UK; liuxiaochuan2020@xjtu.edu.cn (X.L.); bozhou.di20@imperial.ac.uk (B.D.); h.liu19@imperial.ac.uk (H.L.); saksham.dhawan12@imperial.ac.uk (S.D.); mkopec@ippt.pan.pl (M.K.); liliang.wang@imperial.ac.uk (L.W.); 2School of Mechanical Engineering, Xi’an Jiao Tong University, Xi’an 710049, China; 3Department of Mechanical and Manufacturing Engineering, University of Cyprus, Nicosia 1678, Cyprus; politis.denis@ucy.ac.cy; 4Institute of Fundamental Technological Research Polish Academy of Sciences, Pawinskiego 5B, 02-106 Warsaw, Poland

**Keywords:** formability, M-K model, failure criteria, composite sandwich panel

## Abstract

In the present work, the compatibility relationship on the failure criteria between aluminium and polymer was established, and a mechanics-based model for a three-layered sandwich panel was developed based on the M-K model to predict its Forming Limit Diagram (FLD). A case study for a sandwich panel consisting of face layers from AA5754 aluminium alloy and a core layer from polyvinylidene difluoride (PVDF) was subsequently conducted, suggesting that the loading path of aluminium was linear and independent of the punch radius, while the risk for failure of PVDF increased with a decreasing radius and an increasing strain ratio. Therefore, the developed formability model would be conducive to the safety evaluation on the plastic forming and critical failure of composite sandwich panels.

## 1. Introduction

The Forming Limit Diagram (FLD) is a mature method to describe the formability of sheet metals. The right side of the FLD, representing positive minor and major strain, was proposed by Keeler and Backofen [1] and subsequently extended by Goodwin [2] with the addition of the negative minor strain. As the metals exhibit different behaviour at various strain rates and temperatures, multiple Forming Limit Curves (FLCs) are generally presented on a single FLD, of which biaxial strain states could represent necking and subsequent failure.

The Nakajima test is one of the most commonly used methods to determine the FLD, and has been defined in standards such as ISO 12004-2 [3] and BSI 12004-2 [4], in which an upper and lower die with draw beads are applied to ensure the flow of the sheet metal along the perimeter of the hole, and a hemispherical punch is used to plastically deform the material. Sheet metal samples are designed with different widths, which leads to a varied strain ratio between the lateral direction (minor strain) and longitudinal direction (major strain). In order to demonstrate a linear strain ratio evolution (loading path), a lubricant is used to minimise the friction. Consequently, the samples with different strain ratios are able to capture linear loading paths, and their critical strains at the verge of necking could be recorded on FLCs. The FLD is subsequently generated by obtaining multiple critical strains from different sample designs.

Theoretical FLD prediction models have been the subject of significant research, with the development of models including the Swift’s diffuse necking model [5], Hill’s localized necking model [6], the bifurcation analysis-based models [7] and the Marciniake-Kuczynski (M-K) model [8]. Especially, the M-K model enabled the prediction of forming limits for various metals, such as 2060 aluminium alloy [9], 7075 aluminium alloy [10], DP1180 steel [11] and DC01 low-carbon steel [12] at different loading paths, strain rates and elevated temperatures. When combined with the rate-dependent plasticity, polycrystal, self-consistent (VPSC) model, the model was able to realistically predict the fracture behaviour of metals [13]. The fracture angle and local inhomogeneous deformation bands could also be predicted by considering the effect of plastic anisotropy into the model [14]. In addition to the FLD, a generalised forming limit diagram (GFLD) allowing proportional loading with all six independent stress components could be obtained by using the M-K model [15], which has been applied to the prediction of the mesoscaled deformation of clad foils [16].

In recent years, sandwich panels bonded by a polymer as the core layer and metals as the skin layers are being increasingly applied to the construction [17,18,19], aircraft [20,21] and automobile sectors [22] due to the excellent strength/stiffness-to-weight ratio, thermal insulation, bearing capacity and cost-effectiveness [23]. Among them, sandwich panels made from aluminium alloys and polyvinylidene difluoride (PVDF) have found particular applications on lightweight sealing and insulating components due to PVDF’s strong chemical resistance, oxidation resistance and shock resistance. Moreover, the range of applications increased by the inclusion of multiwalled carbon nanotubes (MWCNT) reinforced honeycomb structures to enhance the damping effect [24,25]. At present, a great challenge on the application of the sandwich panels is that the core polymer layer is found to fail earlier than the skin metal layer due to its low resistance to normal pressure. In order to increase the formability of polymer layers, polyethylene [26], polymethyl-methacrylate (PMMA) [27] and polypropylene-polyethylene [28] were applied as adhesion layers. In addition, a density-based topology optimization method was designed and integrated with a multistage algorithm (GA) to optimise the formability of the carbon-fibre-reinforced thermoplastics as the core material of the sandwich panels [29]. To this end, although great efforts have been made to improve the performance of these metal-polymer sandwich panels, the fracture mode and crack propagation of polymers is different to those of metals, leading to lower FLCs and thus earlier failure of the polymer layers [30]. However, the traditional FLD model is not capable of predicting the formability of composite materials, resulting in inaccurate failure prediction of the sandwich panels.

In order to overcome this limitation, the compatibility relationship on the failure criteria between aluminium and polymer layers was established in the present work, and a mechanics-based model for a three-layered sandwich panel was subsequently developed based on the M-K model to predict its formability. Consequently, the critical failures of the sandwich panel at various strain ratios and curvature radii were predicted. Furthermore, the analytical failure solution of the sandwich panel consisting of two AA5754 aluminium alloy layers and a polyvinylidene difluoride (PVDF) layer was presented as a case study, demonstrating the curvature-radius-dependent FLD.

## 2. The Mechanics-Based Analysis of the Nakajima Test with Sandwich Panels

### 2.1. Principles of Plastic Deformation

The Logan-Hosford yield criterion [31] was applied as the principle to describe the anisotropic behaviour of metal layers under the plane stress state:(1)$$R_{2}\sigma_{1}^{l}+R_{1}\sigma_{2}^{l}+R_{1}R_{2}{\left(\sigma_{1}-\sigma_{2}\right)}^{l}=R_{2}(R_{1}+1){\bar{\sigma }}^{l}$$
(2)$$R_{1}{\left(2d\varepsilon_{2}+d\varepsilon_{1}\right)}^{l}+R_{2}{\left(2d\varepsilon_{1}+d\varepsilon_{2}\right)}^{l}+R_{1}R_{2}{\left(d\varepsilon_{1}-d\varepsilon_{2}\right)}^{l}=R_{2}(R_{1}+1)d{\bar{\varepsilon }}^{l}$$
where $R_{1}$ and $R_{2}$ were the *r*-values in the first and second principal directions, and l was a material constant. If a stress ratio was defined as $\alpha =\sigma_{2}/\sigma_{1}$, Equation (1) could be rewritten as:(3)$$[R_{2}+R_{1}\alpha^{l}+R_{1}R_{2}{\left(1-\alpha \right)}^{l}]\sigma_{1}^{l}=R_{2}(R_{1}+1){\bar{\sigma }}^{l}$$

The external loading path was assumed as linear, leading to a constant strain ratio $\beta =d\varepsilon_{2}/d\varepsilon_{1}=\varepsilon_{2}/\varepsilon_{1}$. Consequently, the equivalent strain was expressed as:(4)$$[R_{1}{\left(2\beta +1\right)}^{l}+R_{2}{\left(\beta +2\right)}^{l}+R_{1}R_{2}{\left(1-\beta \right)}^{l}]\varepsilon_{1}^{l}=R_{2}(R_{1}+1){\bar{\varepsilon }}^{l}$$

The constitutive relationship of the sheet metal was described as a power law between the flow stress and plastic deformation, where K was the strength coefficient and n was the strain-hardening exponent:(5)$$\bar{\sigma }=K{\bar{\varepsilon }}^{n}$$

The associate flow rule was used to describe the plastic flow behavior as Equation (6) by defining a positive scaler dλ, in order to establish the relationship between $d\varepsilon_{ij}$ and $\partial \bar{\sigma }/\partial \sigma_{ij}$. As a result, the strain ratio β could be expressed as a function of the stress ratio α, as shown in Equation (8).
(6)$$d\varepsilon_{ij}=d\lambda \frac{\partial \bar{\sigma }}{\partial \sigma_{ij}}\to d\lambda =d\varepsilon_{ij}/\frac{\partial \bar{\sigma }}{\partial \sigma_{ij}}$$
(7)$$\frac{d\varepsilon_{1}}{R_{2}\sigma_{1}^{l-1}+R_{1}R_{2}{\left(\sigma_{1}-\sigma_{2}\right)}^{l-1}}=\frac{d\varepsilon_{2}}{R_{1}\sigma_{2}^{l-1}-R_{1}R_{2}{\left(\sigma_{1}-\sigma_{2}\right)}^{l-1}}$$
(8)$$\beta =\frac{R_{1}\alpha -R_{1}R_{2}{\left(1-\alpha \right)}^{l-1}}{R_{2}+R_{1}R_{2}{\left(1-\alpha \right)}^{l-1}}$$

### 2.2. Strain and Stress Analysis of Sandwich Panels

In the Nakajima tests, the friction was generally ignored in the derivation under the lubricated conditions. Therefore, the failure would occur at the apex of the dome in the frictionless case. Figure 1 shows the stress state at an infinitesimal apex element in a single metal layer, of which the stress equilibrium along the thickness direction was derived as:(9)$$2\sigma_{1}tRd\theta \sin\frac{d\varphi }{2}+2tRd\varphi \sin\frac{d\theta }{2}=pR^{2}d\theta d\varphi$$
where *R* was the radius of the punch, t was the thickness at the deformed stage, and p was the contact pressure between the sheet and forming tools on the element. Considering that $\sin\frac{d\varphi }{2}\approx \frac{d\varphi }{2}$ and $\sin\frac{d\theta }{2}\approx \frac{d\theta }{2}$ for the infinitesimal elements, the stress equilibrium equation could be simplified as:(10)$$(\sigma_{1}+\alpha \sigma_{1})t=pR$$

The stress equilibrium equation was modified for the three-layered sandwich panel, consisting of two skin metal layers (layer I and III) and one core polymer layer (layer II), as shown in Figure 2. For each layer, Equations (9) and (10) remained valid as long as the contact pressure was amended to the pressure difference between the top and bottom surfaces, considering that the contact pressure decreased from the inner face to the outer face. Thus, the stress equilibrium was developed as Equations (11)–(13):(11)$$(\sigma_{1}^{I}+\alpha^{I}\sigma_{1}^{I})t^{I}=(p_{1}-p_{2})R$$
(12)$$(\sigma_{1}^{II}+\alpha^{II}\sigma_{1}^{II})t^{II}=(p_{2}-p_{3})R$$
(13)$$(\sigma_{1}^{III}+\alpha^{III}\sigma_{1}^{III})t^{III}=(p_{3}-p_{4})R$$
where $\sigma_{1}^{I}$, $\sigma_{1}^{II}$ and $\sigma_{1}^{III}$ were the first principal stresses on each layer, $t^{I}$, $t^{II}$ and $t^{III}$ were the thicknesses of each layer at the deformation stage, and $p_{1}$, $p_{2}$, $p_{3}$ and $p_{4}$ were the contact pressures on each contact surface.

Based on the volume constancy, the relationship between the initial thickness and the first principal strain, $\varepsilon_{1}$ could be derived as:(14)$$\ln\frac{t^{I/II/III}}{t_{0}^{I/II/III}}=\varepsilon_{3}=-\varepsilon_{1}-\varepsilon_{2}=-(1+\beta )\varepsilon_{1}$$
where $t_{0}^{I/II/III}$ was the initial thickness of each layer. Note that the strain components and strain ratio β in the three layers were the same, due to the compatibility.

By eliminating $t^{I}$, $t^{II}$ and $t^{III}$ in Equations (11)–(14), the relationship of strains was simplified as:(15)$$R\cdot \exp[(1+\beta )\varepsilon_{1}]=\frac{t_{0}^{I}\sigma_{1}^{I}(1+\alpha^{I})}{p_{1}-p_{2}}=\frac{t_{0}^{II}\sigma_{1}^{II}(1+\alpha^{II})}{p_{2}-p_{3}}=\frac{t_{0}^{II}\sigma_{1}^{III}(1+\alpha^{III})}{p_{3}-p_{4}}$$

### 2.3. Failure Criteria of Polymer Layer

A critical normal pressure $p_{cp}$ was defined as the failure criteria of the polymer layer. Thus, the boundary conditions at failure were the contact pressure $p_{2}=p_{cp}$ applied on the polymer layer and $p_{4}=0$ applied on the skin layer. As a result, Equation (15) was rewritten as:(16)$$R\cdot \exp[(1+\beta )\varepsilon_{1}]=\frac{t_{0}^{I}\sigma_{1}^{I}(1+\alpha^{I})}{p_{1}-p_{cp}}=\frac{t_{0}^{II}\sigma_{1}^{II}(1+\alpha^{II})}{p_{cp}-p_{3}}=\frac{t_{0}^{III}\sigma_{1}^{III}(1+\alpha^{III})}{p_{3}}$$

Combining Equations (3) and (4), the power law of Equation (5) was modified as below:(17)$${\left[\frac{R_{2}+R_{1}\alpha^{l}+R_{1}R_{2}{\left(\alpha -1\right)}^{l}}{R_{2}(R_{1}+1)}\right]}^{1/l}\sigma_{1}=K{\left[\frac{R_{2}{\left(\beta +2\right)}^{l}+R_{1}{\left(2\beta +1\right)}^{l}+R_{1}R_{2}{\left(\beta -1\right)}^{l}}{R_{2}(R_{1}+1)}\right]}^{n/l}\varepsilon_{1}^{n}$$

By eliminating $\sigma_{1}^{I/II/III}$ in Equation (16),
(18)$$\frac{R\cdot \exp[(1+\beta )\varepsilon_{1}]}{1+\alpha^{I}}=\frac{t_{0}^{I}K^{I}\varepsilon_{1}^{n}{\left[\frac{R_{2}^{I}{\left(\beta +2\right)}^{l}{}^{{}^{I}}+R_{1}^{I}{\left(2\beta +1\right)}^{l}{}^{{}^{I}}+R_{1}^{I}R_{2}^{I}{\left(\beta -1\right)}^{l}{}^{{}^{I}}}{R_{2}^{I}(R_{1}^{I}+1)}\right]}^{n^{I}/l^{I}}}{(p_{1}-p_{cp}){\left[\frac{R_{2}^{I}+R_{1}^{I}\alpha^{l}{}^{{}^{I}}+R_{1}^{I}R_{2}^{I}{\left(\alpha -1\right)}^{l^{I}}}{R_{2}^{I}(R_{1}^{I}+1)}\right]}^{1/l^{I}}}$$
(19)$$\frac{R\cdot \exp[(1+\beta )\varepsilon_{1}]}{1+\alpha^{II}}=\frac{t_{0}^{II}K^{II}\varepsilon_{1}^{n}{\left[\frac{R_{2}^{II}{\left(\beta +2\right)}^{l}{}^{{}^{II}}+R_{1}^{II}{\left(2\beta +1\right)}^{l}{}^{{}^{II}}+R_{1}^{II}R_{2}^{II}{\left(\beta -1\right)}^{l}{}^{{}^{II}}}{R_{2}^{II}(R_{1}^{II}+1)}\right]}^{n^{II}/l^{II}}}{(p_{cp}-p_{3}){\left[\frac{R_{2}^{II}+R_{1}^{II}\alpha^{l}{}^{{}^{II}}+R_{1}^{II}R_{2}^{II}{\left(\alpha -1\right)}^{l^{II}}}{R_{2}^{II}(R_{1}^{II}+1)}\right]}^{1/l^{II}}}$$
(20)$$\frac{R\cdot \exp[(1+\beta )\varepsilon_{1}]}{1+\alpha^{III}}=\frac{t_{0}^{III}K^{III}\varepsilon_{1}^{n}{\left[\frac{R_{2}^{III}{\left(\beta +2\right)}^{l}{}^{{}^{III}}+R_{1}^{III}{\left(2\beta +1\right)}^{l}{}^{{}^{III}}+R_{1}^{III}R_{2}^{III}{\left(\beta -1\right)}^{l}{}^{{}^{III}}}{R_{2}^{III}(R_{1}^{III}+1)}\right]}^{n^{III}/l^{III}}}{p_{3}{\left[\frac{R_{2}^{III}+R_{1}^{III}\alpha^{l}{}^{{}^{III}}+R_{1}^{III}R_{2}^{III}{\left(\alpha -1\right)}^{l^{III}}}{R_{2}^{III}(R_{1}^{III}+1)}\right]}^{1/l^{III}}}$$

By eliminating $p_{3}$ in Equations (19) and (20),


(21)
$$\begin{array}{l} p_{cp}\cdot R\cdot \exp[(1+\beta )\varepsilon_{1}]=\\ \frac{t_{0}^{II}K^{II}\varepsilon_{1}^{n}{\left[\frac{R_{2}^{II}{\left(\beta +2\right)}^{l}{}^{{}^{II}}+R_{1}^{II}{\left(2\beta +1\right)}^{l}{}^{{}^{II}}+R_{1}^{II}R_{2}^{II}{\left(\beta -1\right)}^{l}{}^{{}^{II}}}{R_{2}^{II}(R_{1}^{II}+1)}\right]}^{n^{II}/l^{II}}\cdot (1+\alpha^{II})}{{\left[\frac{R_{2}^{II}+R_{1}^{II}\alpha^{l}{}^{{}^{II}}+R_{1}^{II}R_{2}^{II}{\left(\alpha -1\right)}^{l}{}^{{}^{II}}}{R_{2}^{II}(R_{1}^{II}+1)}\right]}^{1/l^{II}}}\\ +\frac{t_{0}^{III}K^{III}\varepsilon_{1}^{n}{\left[\frac{R_{2}^{III}{\left(\beta +2\right)}^{l}{}^{{}^{III}}+R_{1}^{III}{\left(2\beta +1\right)}^{l}{}^{{}^{III}}+R_{1}^{III}R_{2}^{III}{\left(\beta -1\right)}^{l}{}^{{}^{III}}}{R_{2}^{III}(R_{1}^{III}+1)}\right]}^{n^{III}/l^{III}}\cdot (1+\alpha^{III})}{{\left[\frac{R_{2}^{III}+R_{1}^{II}\alpha^{l}{}^{{}^{III}}+R_{1}^{III}R_{2}^{III}{\left(\alpha -1\right)}^{l}{}^{{}^{III}}}{R_{2}^{III}(R_{1}^{III}+1)}\right]}^{1/l^{III}}} \end{array}$$


If the strain ratio β and other material constants were given, $\alpha^{II}$ and $\alpha^{III}$ could be calculated by Equation (8), resulting in the only unknown factor, $\varepsilon_{1}$ in Equation (21), which would be solved numerically. It should be noted that the solution might not exist, due to the fact that the term on the left side of Equation (21) (Term 1) increased exponentially with increasing $\varepsilon_{1}$, while the term on the right side (Term 2) increased polynomially. This indicated that Term 1 increased at a higher rate than that of Term 2 after a given point, where the derivative of each side was equal. Thus, there were four bifurcation conditions, as shown in Figure 3:(a)If Term 1 was always larger than Term 2, no solution existed, suggesting that the sandwich panel would not fail as the failure criteria of the polymer layer was not met;(b)If Term 1 was larger than Term 2 when $\varepsilon_{1}=0$ and the two terms became equal at a given point, where the derivative of each was equal, a unique solution existed;(c)If Term 1 was larger than Term 2 when $\varepsilon_{1}=0$ and it became smaller at a given point, where the derivative of each was equal, two solutions existed, of which the smaller one was true as the failure occurred and the increase in $\varepsilon_{1}$ terminated;(d)If Term 1 was smaller than Term 2 when $\varepsilon_{1}=0$, a unique solution existed.

It was advised to limit the solution domain to be positive with an initial estimate of 0 when solving Equation (21). Once $\varepsilon_{1}$ was solved, the pressure on each interface could be obtained by using Equations (18)–(20).

### 2.4. Failure Criteria of Metal Layers

The M-K model [8] was applied to determine the forming limit of the metal layers, assuming that a perfection Zone *a* and an imperfection Zone *b* coexisted in the metal layers, while a perfection Zone *a* existed in the polymer layer, as shown in Figure 4. Once the failure criterion of Equation (22) was met, the metal layers failed.
(22)$$\frac{d\varepsilon_{3b}^{I/III}}{d\varepsilon_{3a}^{I/III}}=\frac{\varepsilon_{3b,i}-\varepsilon_{3b,i-1}}{\varepsilon_{3a,i}-\varepsilon_{3a,i-1}}>10$$

In order to activate the failure criteria of the metal layers, an initial geometrical nonhomogeneity along the thickness direction, known as the ‘imperfection factor f’, was defined as Equation (23), which had evolved from the initial imperfection factor $f_{0}$. The upper script I/III was neglected to simplify the derivation of the M-K model.
(23)$$f=\frac{t_{b}}{t_{a}},f_{0}=\frac{t_{b0}}{t_{a0}}$$
where $t_{a}$ and $t_{b}$ were the instantaneous thicknesses of Zones *a* and *b*, while $t_{a0}$ and $t_{b0}$ were the initial thickness of Zones *a* and *b.* The relationship between the imperfection factor and the initial imperfection factor was derived as:(24)$$\varepsilon_{3a}=\ln(\frac{t_{a}}{t_{a0}})\Rightarrow t_{a}=t_{a0}\exp(\varepsilon_{3a})$$
(25)$$\varepsilon_{3b}=\ln(\frac{t_{b}}{t_{b0}})\Rightarrow t_{b}=t_{b0}\exp(\varepsilon_{3b})$$
(26)$$f=f_{0}\exp(\varepsilon_{3b}-\varepsilon_{3a})$$

The application of the external loading on Zone *a* led to a constant stress ratio $\alpha_{a}$ and strain ratio $\beta_{a}$. $\varepsilon_{1a}$ was assumed to grow by dε at any step, $\varepsilon_{2a}$ could be calculated based on the linear loading assumption and $\varepsilon_{3a}$ could be derived considering the volume constancy:(27)$$\varepsilon_{1a,i}=\varepsilon_{1a,i-1}+d\varepsilon$$
(28)$$\varepsilon_{2a,i}=\beta_{a}\varepsilon_{1a,i}$$
(29)$$\varepsilon_{3a,i}=-(\varepsilon_{1a,i}+\varepsilon_{2a,i})=-(1+\beta_{a})\varepsilon_{1a,i}$$

The first principal stress could be calculated from the modified power law in Equation (17):(30)$$\sigma_{ia,i}=\frac{K{\left[\frac{R_{2}{\left(\beta_{a}+2\right)}^{l}+R_{1}{\left(2\beta_{a}+1\right)}^{l}+R_{1}R_{2}{\left(\beta_{a}-1\right)}^{l}}{R_{2}(R_{1}+1)}\right]}^{n/l}}{{\left[\frac{R_{2}+R_{1}\alpha^{l}+R_{1}R_{2}{\left(\alpha -1\right)}^{l}}{R_{2}(R_{1}+1)}\right]}^{1/l}}\cdot \varepsilon_{1a,i}^{n}$$

Subsequently, $\sigma_{2a}$ could be expressed by using the stress ratio in Equation (31), and $\sigma_{3a}$ was assumed to be 0 for metal sheets.
(31)$$\sigma_{2a,i}=\alpha_{a}\sigma_{1a,i}$$

The stress and strain states on Zone *b* were predicted considering the compatibility of strain and equilibrium of stress between the two zones:(32)$$\varepsilon_{2b,i}=\varepsilon_{2a,i}$$
(33)$$\sigma_{1b,i}=\frac{1}{f}\sigma_{1a,i}=-\frac{\sigma_{1a,i}}{f_{0}\exp(\varepsilon_{2b,i}/\beta_{b,i}+\varepsilon_{2b,i}+\varepsilon_{3a,i})}$$

Note that the loading path of Zone *b* was not linear but followed the compatibility relationship with Zone *a*; thus, $\beta_{b,i}$ was changed for a different step *i*.

The plastic deformation of Zone *b* could be modelled based on the modified power law of Equation (17):(34)$$\begin{array}{l} {\left[\frac{R_{2}+R_{1}\alpha_{b,i}^{l}+R_{1}R_{2}{\left(\alpha_{b,i}-1\right)}^{l}}{R_{2}(R_{1}+1)}\right]}^{1/l}\cdot \frac{\sigma_{1a,i}}{f_{0}\exp(\varepsilon_{2b,i}/\beta_{b,i}+\varepsilon_{2b,i}+\varepsilon_{3a,i})}\\ +K{\left[\frac{R_{2}{\left(\beta_{b,i}+2\right)}^{l}+R_{1}{\left(2\beta_{b,i}+1\right)}^{l}+R_{1}R_{2}{\left(\beta_{b,i}-1\right)}^{l}}{R_{2}(R_{1}+1)}\right]}^{n/l}\cdot {\left(\frac{\varepsilon_{2b,i}}{\beta_{b,i}}\right)}^{n}=0 \end{array}$$

From Equation (8), the strain ratio on Zone *b* was expressed as:(35)$$\beta_{b,i}=\frac{R_{1}\alpha_{b,i}-R_{1}R_{2}{\left(1-\alpha_{b,i}\right)}^{l-1}}{R_{2}+R_{1}R_{2}{\left(1-\alpha_{b,i}\right)}^{l-1}}$$

By substituting Equation (35) into Equation (34), $\alpha_{b,i}$ remained the only unknown in Equation (34). After solving for $\alpha_{b,i}$ from Equation (34), all the stress and strain components on Zone *b* could be derived accordingly. In order to numerically solve the model constants, it was necessary to define $\varepsilon_{1c}^{I/II/III}$ as the limit major strain of each layer. Specifically, $\varepsilon_{1c}$ and $\varepsilon_{2c}$ had to be defined as the limit major and minor strain of the sandwich panel. The flow chart of the analytical model is shown in Figure 5.

## 3. Case Study

In the case study that follows, the model was applied to a sandwich panel, which was produced by combining two skin layers of AA5754 aluminium alloy with a thickness of 1 mm and one core layer of polyvinylidene difluoride (PVDF) with a thickness of 0.1 mm. The properties of AA5754 and PVDF are presented in Table 1 [32,33]. The imperfection factor of AA5754 was set to 0.95. The failure prediction was conducted at a punch radius range between 80 to 180 mm.

Figure 6 shows the FLCs of PVDF and AA5754 under different punch radii. The blue curves represent the critical failure of the AA5754 layers (since the properties of the two aluminium layers were the same and thus the results coincided) and followed a typical V-shape, indicating that the loading path was linear and independent of the punch radius. The green curves represent the critical failure of the PVDF layer, which was significantly dependent on the punch radius. Specifically, when the punch radius was small (*R* = 80 or 100 mm), the FLC of PVDF was much lower than that of AA5754. However, the FLC of PVDF increased with an increasing radius. When the radius reached 120 mm, a part of the FLC was greater than that of AA5754, suggesting that the PVDF layer would not fail before the AA5754 layer. Meanwhile, the critical failure of PVDF was not solved by the model at some low strain-ratio values, indicating that the polymer would not fail under those conditions. When the radius increased to 180 mm, no critical failure of PVDF was predicted at all, suggesting that the entire layer was safe regardless of the strain state. The FLCs of PVDF and AA5754 under different punch radii were combined, as shown in Figure 7, to generate a curvature-radius-dependent FLD. As can be seen, it was found that the risk for failure of PVDF increased with a decreasing radius and an increasing strain ratio.

It should be noted that each FLC in Figure 6 could be used to determine the formability of a sandwich panel in a Nakajima test under certain punch radii and strain ratios. However, different regions of a complex-shaped component may withstand various punch radii and strain ratios, which may even vary during forming. Therefore, it is necessary to combine the developed model and a finite element method to predict the evolutionary formability of each element when forming complex-shaped sandwich panels [24,25]. In addition, the developed model has the potential capability of predicting the formability of sandwich panels made from other composite materials with multiple layers by using the proposed compatibility relationship of the failure criteria between polymers and metals.

## 4. Conclusions

In the present research, an analytical formability model for the sandwich panel has been developed and its capability has been demonstrated by predicting the critical failure of a sandwich panel consisting of two skin AA5754 layers and a core PVDF layer as a case study. The main conclusions from the work were summarised as follows:The failure of the polymer layer was curvature-radius dependent. At a small radius, the polymer layer readily failed, while at a large radius, the sandwich panel would be more formable. This feature contributed to the curvature-radius-dependent FLD of the sandwich panel.Under the same punch radius/curvature, the polymer layer always failed rapidly at a larger strain ratio. The worst case was the equi-biaxial (β = 1) loading path.The developed FLD model overcame the limitation of traditional FLD models and was capable of predicting the formability of sandwich panels made from composite materials.

The work in this study provided a safety evaluation and theoretical guidance on the plastic forming and critical failure of composite sandwich panels for lightweight sealing and insulating component applications. It is recommended that subsequent studies conduct Nakajima tests and forming trials of complex-shaped components to experimentally verify the formability of the applied composite material and validate the developed model proposed in this study.

## Figures and Tables

**Figure 1 materials-15-04140-f001:**
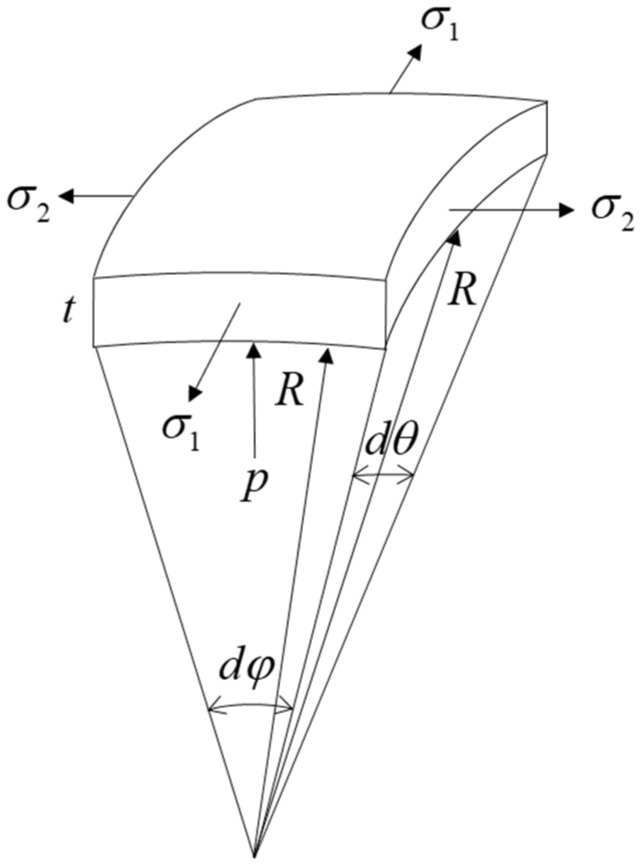
Stress at an infinitesimal apex element in a single metal layer.

**Figure 2 materials-15-04140-f002:**
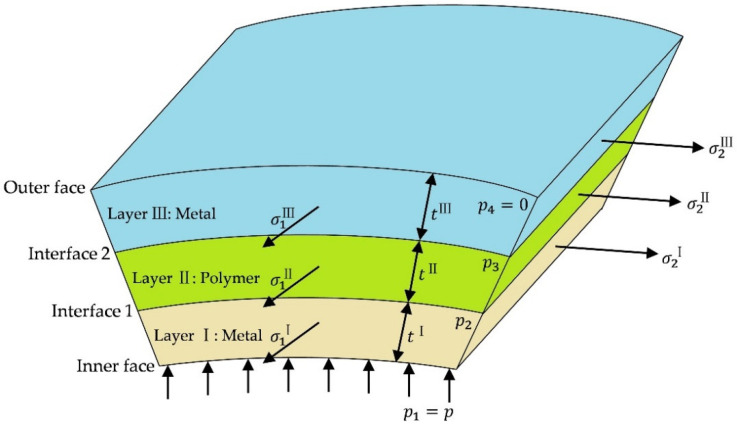
Structure of the three-layered sandwich panel.

**Figure 3 materials-15-04140-f003:**
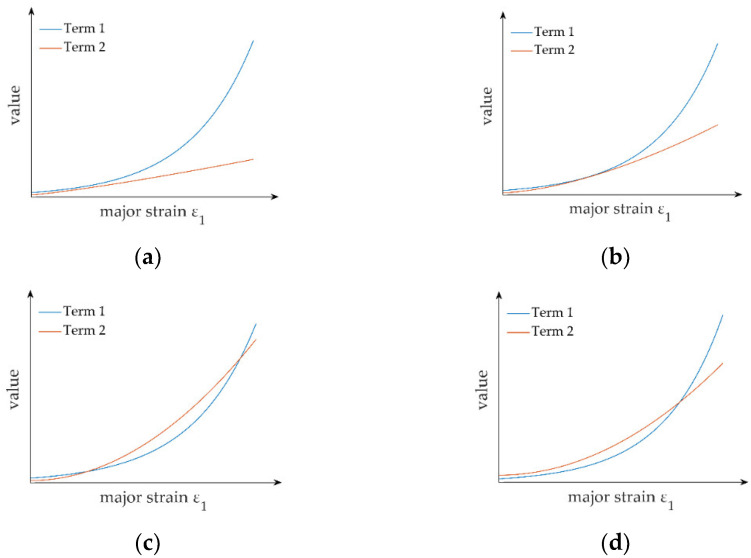
Bifurcation conditions of the failure criteria of the polymer layer: (**a**) Term 1 was larger than Term 2; (**b**) Term 1 was initially larger than Term 2 and became equal at a given point; (**c**) Term 1 was initially larger than Term 2 and became smaller at a given point; (**d**) Term 1 was smaller than Term 2.

**Figure 4 materials-15-04140-f004:**
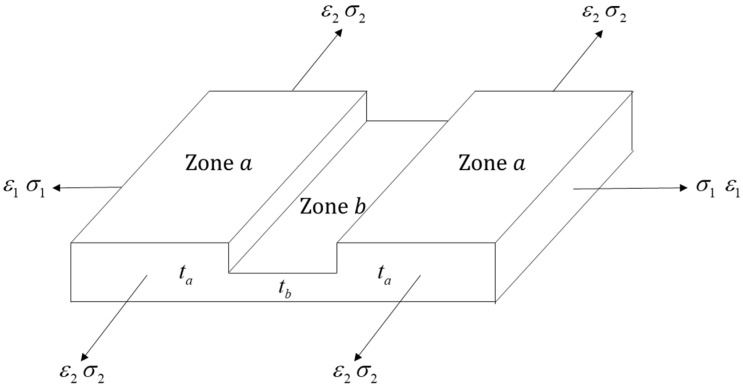
Schematic diagram of Zone *a* and *b* in the M-K model.

**Figure 5 materials-15-04140-f005:**
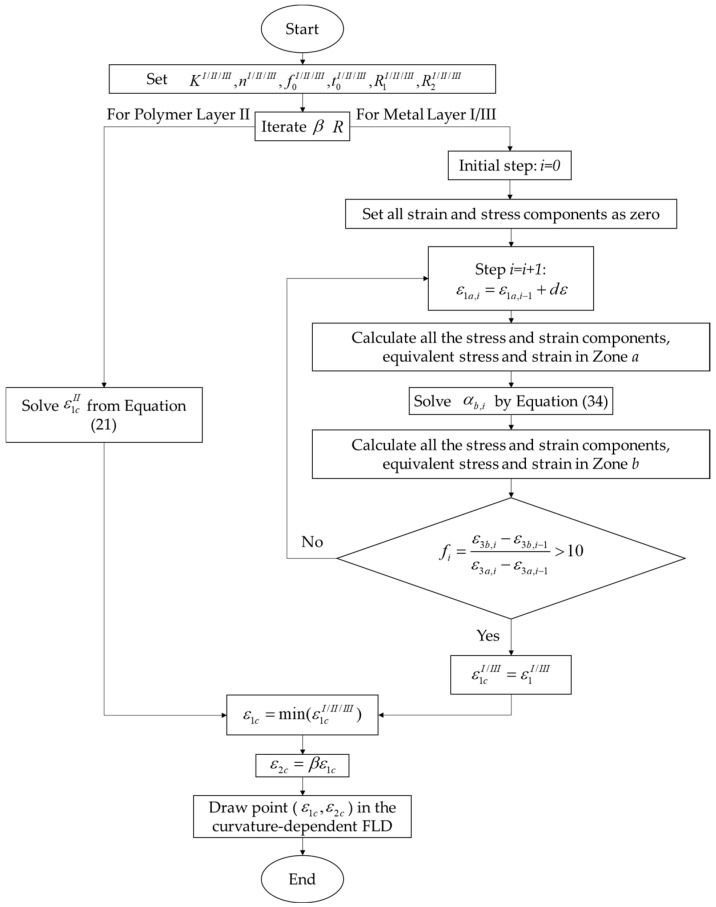
Flow chart of the model.

**Figure 6 materials-15-04140-f006:**
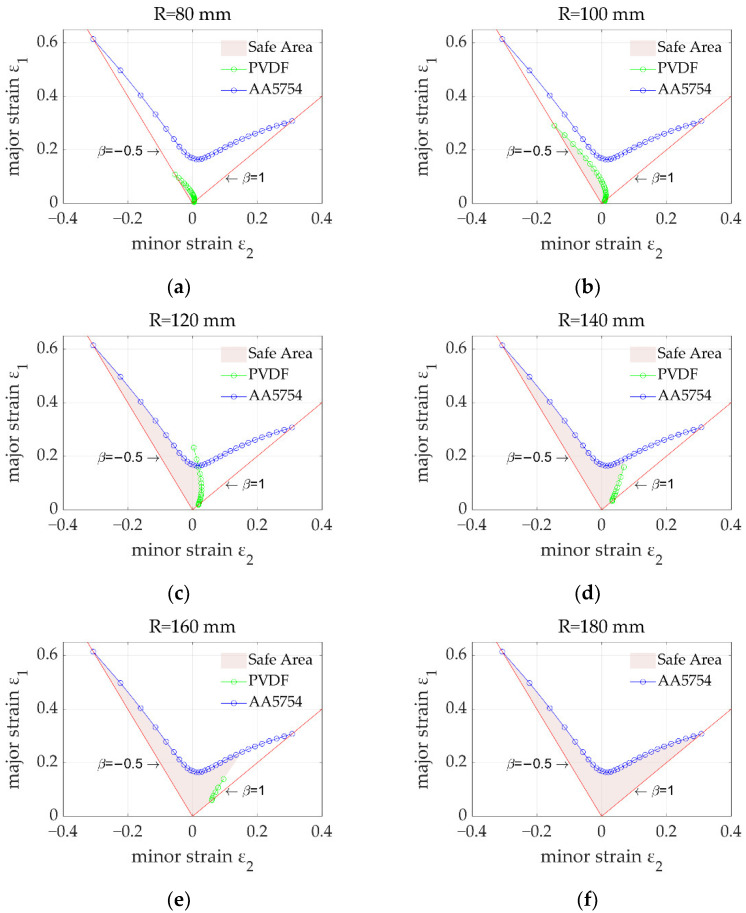
FLCs of PVDF and AA5754 with different punch radii: (**a**) 80 mm; (**b**) 100 mm; (**c**) 120 mm; (**d**) 140 mm; (**e**) 160 mm; (**f**) 180 mm.

**Figure 7 materials-15-04140-f007:**
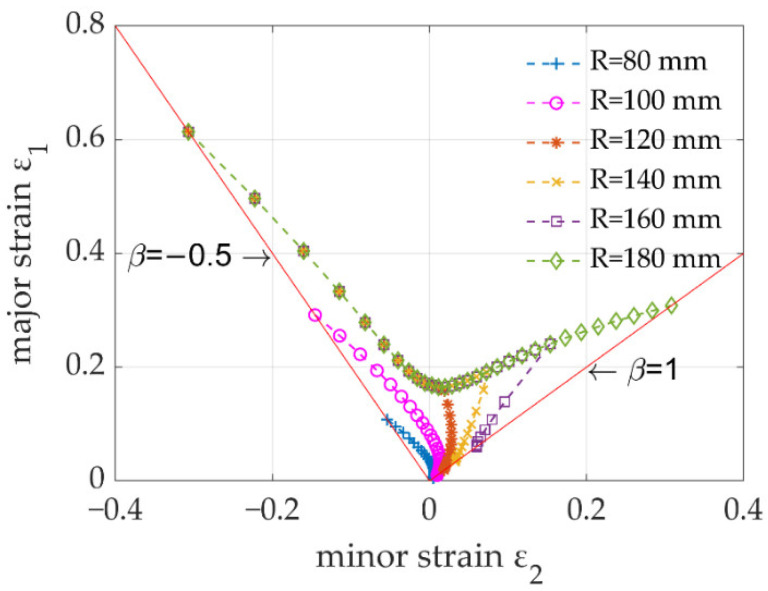
Curvature-radius-dependent FLD for the sandwich panel.

**Table 1 materials-15-04140-t001:** Mechanical properties of AA5754 and PVDF.

Material	$R_{1}$	$R_{2}$	$\sigma_{0}\;[\mathrm{MPa}]$	*K* [MPa]	*n*	*l*
AA5754	0.73	0.69	0	474	0.317	8
PVDF	1	1	0	6.51	0.465	2

## Data Availability

Data is contained within the article.

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
