# Peer review of "Development of a Formability Prediction Model for Aluminium Sandwich Panels with Polymer Core"

_materials, 2022, doi:10.3390/ma15124140_

Round 1
Reviewer 1 Report
The work is interesting. Nevertheless, there are issues that need to be clarified:
1. Please indicate the potential applications of the sandwich panels consisting of AA5754 aluminium alloy and polyvinylidene difluoride (PVDF).
2. In the Figure 2, there are two values R5 and R8? Why?
3. Please explain "l" (lambda) in equation (6).
4. Line 120: it should be: “… Eq. (15) was rewritten as:”
5. Please check the correctness of the formulas: (15), (16), (20) and (21).
6. Please correct the legend in figure 7.
7. Which of the curves from Figure 8 should be used when analysing the process of forming the complex parts made of the sandwich panel?
8. Can the proposed method of calculating the Forming Limit Diagram (FLD) be applied to other types of composite materials, e.g. materials consisting of more layers?
9. Do you verify the obtained results experimentally? and how?
10. Please indicate clearly what is the novelty of the work compared to the research conducted so far.
Reviewer 2 Report
1. Mention the materials used for face and core layer of sandwich panels in abstract.
2. Modify the abstract 1st -problem, 2nd -your approach and 3rd-your findings
3. Introduction can be improved and elaborate about formability and specific applications of sandwich panels
4. Based on the literature review presented it is not possible to identify the novelty of the presented work. What is the advantage of using the presented formulation?
5. Literature is very less. Cite more recent works on sandwich panels
6. Authors are focusing on characteristics of sandwich panels and in this regard, they are invited to strengthen the literature review by incorporating the following literature: https://doi.org/10.1016/j.compstruct.2021.114048; https://doi.org/10.1080/15397734.2021.1950550;
7. There is no information on the AA5754 aluminium alloy face sheet and polyvinylidene difluoride core material properties (youngs modulus, shear modulus, density, poisson ratio) used in the analysis.
8. Some text are not visible in figure 7
9. There is no validation about presented model. As the sandwich panels used in this work, the validation must be done.
10. Based on the analysis and discussion of numerical simulation, can experiments be considered for verification?
Round 2
Reviewer 2 Report
1. Modify the word “two face layers” to face layers in the abstract.
2. Remove the 1st four lines from abstract. Start the abstract with “In the present study”
3. Change the title like this “Development of a Formability Prediction Model for Aluminium Sandwich Panels with polyvinylidene difluoride core”
